# Lifetime Prediction Methods for Degradable Polymeric Materials—A Short Review

**DOI:** 10.3390/ma13204507

**Published:** 2020-10-12

**Authors:** Angelika Plota, Anna Masek

**Affiliations:** Institute of Polymer and Dye Technology, Faculty of Chemistry, Lodz University of Technology, Stefanowskiego 12/16, 90-924 Lodz, Poland; angelika.plota@gmail.com

**Keywords:** lifetime, degradation, accelerated aging, polymer, kinetic models, thermal analysis

## Abstract

The determination of the secure working life of polymeric materials is essential for their successful application in the packaging, medicine, engineering and consumer goods industries. An understanding of the chemical and physical changes in the structure of different polymers when exposed to long-term external factors (e.g., heat, ozone, oxygen, UV radiation, light radiation, chemical substances, water vapour) has provided a model for examining their ultimate lifetime by not only stabilization of the polymer, but also accelerating the degradation reactions. This paper presents an overview of the latest accounts on the impact of the most common environmental factors on the degradation processes of polymeric materials, and some examples of shelf life of rubber products are given. Additionally, the methods of lifetime prediction of degradable polymers using accelerated ageing tests and methods for extrapolation of data from induced thermal degradation are described: the Arrhenius model, time–temperature superposition (TTSP), the Williams–Landel–Ferry (WLF) model and 5 isoconversional approaches: Friedman’s, Ozawa–Flynn–Wall (OFW), the OFW method corrected by N. Sbirrazzuoli et al., the Kissinger–Akahira–Sunose (KAS) algorithm, and the advanced isoconversional method by S. Vyazovkin. Examples of applications in recent years are given.

## 1. Introduction

Nowadays, polymeric materials play a significant role in the development of modern civilization and are increasingly replacing traditional materials in almost every field, e.g., packaging, construction and in many other industries. Over the past several decades, the advance of plastics has made for economic development and brought huge benefits to our lives. The average annual production of these materials was only 1.5 million tons in the 1950s and had precipitously increased to nearly 400 million tons in 2018. Plastics are characterized by good durability, easy processability, light weight and low production costs for large, ready-made products [1].

The basic groups of application of polymeric materials in technology result from their specific functional properties. Most often they are used in the following areas: construction [2,3], medicine [4], agriculture [5], food [6], household [7], automotive [8,9] and chemical [10]. The application of plastic as packaging items is especially important, with around 40 million tonnes of plastic film produced from polyethylene alone [11].

Over many years of use, polymeric materials should exhibit good mechanical, physical and chemical properties and maintain their aesthetic qualities. However, during operation over time, they often change their original performance characteristics. The physical properties of elastomer products undergo various changes, as a result of which these products may become useless due to excessive hardening [12], softening [13], cracking [14] or other surface damage [15]. The reason is the susceptibility of polymers to oxidation and degradation processes. In outdoor applications, all polymeric materials degrade eventually [16].

Lifetime prediction of polymeric materials offers the benefit of isolating and identifying material failures against the fallouts of damage resulting in catastrophic harm. Therefore, accurate prediction of the service life of a material is very important in terms of safety, especially considering elastomeric construction materials (e.g., aircraft components, defense applications, nuclear reactor safety construction components) [15,17,18]. Additionally, this is a crucial factor for reliable use of polymers in medicine, engineering or consumer goods production [11].

This manuscript presents an overview of the latest accounts on the impact of the most common environmental factors on the degradation processes of polymeric materials. First, some general ways in which polymers environmentally degrade are set out, describing the factors, mechanisms and various changes caused by these processes. Then, some examples of shelf life for different polymeric materials and some generalities on accelerated aging are presented. The last part concentrates on different methods for lifetime prediction of degradable polymers using kinetic models from the literature for extrapolation of data from induced thermal degradation with examples of applications reported in recent years. The main goal of this work is to increase people’s awareness of the importance of lifetime prediction for different materials due to their safety, influence on the environment and implications for long-term use. Based on polymer ageing research experience and a literature review, this paper provides a new perspective on this subject.

## 2. Degradation of Polymers

Three basic processes of polymer chain scission leading to a reduction in the molecular weight of polymers are known: depolymerization, destruction and degradation [19,20,21]. The depolymerization reaction consists of the thermal decomposition of the polymer to the monomer [19,20,21]. The destruction process represents the decomposition of polymer chains with separation of low molecular weight compounds other than the monomer [22]. The degradation process is the partial decomposition of the polymer, not into low molecular weight products, but into fragments with large but smaller molecular weights than the original polymer, and this is the most common process during the lifetime of a polymeric material [23,24,25,26].

### 2.1. Changes Caused by Degradation

Degradation is a process of structural changes that may result from physical or chemical transformations taking place in polymeric materials under the influence of long-term external factors (e.g., heat, ozone, oxygen, UV radiation, light radiation, chemical substances, water vapour, high energy radiation, dynamic stresses) which cause deterioration of the primary use properties [27,28,29]. As a result, primary properties are lost, and the first visible undesired signs are: a change in colour (e.g., yellowing), gloss (tarnishing) and texture [30,31]. In practice, there are much more complex systems of factors causing material degradation, which often makes it very difficult to distinguish which of these factors has a dominant influence because they act simultaneously [29].

The degradation process most often causes irreversible changes in the polymer, which are the result of such reactions as: crosslinking, chain cutting, thermal oxidation and even destruction [32]. The scheme of changes caused by the degradation process is shown in Figure 1.

Generally, this process can be carried out in an intentional and controlled way, thus gaining practical significance (improving processing operations or recovering mers from polymers) or it is uncontrolled and limits their practical application [29]. There are cases in which, in the first phase of material degradation, the factor causing the process has a positive effect on improving some properties (e.g., mechanical strength) by additional crosslinking of the elastomer under the influence of heat. However, in later stages of the degradation, the progress of ongoing processes, such as excessive cross linking or molecular weight reduction, begin to adversely affect the properties [33].

### 2.2. Types of Degradation

There are different types of polymer degradation such as thermal, photo, hydrolitic, bio and mechanical degradation [24]. Polymeric materials are susceptible to degradation to varying degrees. Degradation initiators and corresponding types of degradation are shown in Figure 2.

Of all the factors that affect the deterioration of elastomers, ozone, oxygen and elevated temperature are the most important. This is called ozone and thermo-oxidative aging [15]. There are strong links between the types of degradation presented above and, as mentioned, there are often several types of degradation at the same time [34]. An example is the simultaneous action of oxygen, light and other atmospheric factors [32].

The effect of the above factors also depends on the duration of their action (exposure time) and the type of polymer tested (including its molecular structure and its defects), the type and content of impurities, as well as the thickness and shape of the product [31].

Studies conducted so far on the process of polymer and plastic degradation show that the susceptibility of polymers depends to a large extent on the degree of crystallinity. Crystalline polymers are more resistant than amorphous ones. Additionally, polymers with a linear structure are more rapidly degraded than branched ones [35]. The polymer’s decomposition also depends on its molecular weight, the larger the molecular weight, the slower the decomposition [36]. The last factor speeding up the degradation process is specific chemical groups. Amide, ester and urea groups are responsible for accelerating the degradation of a polymer because they easily hydrolyze [37].

### 2.3. General Mechanism of Thermal Degradation

Each type of polymer degradation is characterized by a specific molecular mechanism. Additionally, different mechanisms may be involved in the degradation of a material simultaneously [34]. The initial reaction during this process is always cracking of a bond in the polymer chain or other molecule that initiates degradation. A number of secondary reactions can occur as a result of bond breakage, which initiate further bond breakage, recombination or substitution [38].

The decomposition mechanism of polymeric materials is started by an initiation stage, during which free radicals are produced and hydrogen atoms are disconnected by energy from any source: radiation, light, heat or by the presence of an initiator [39], Equation (1):(1)$$R-H\overset{\mathrm{heat},\;\mathrm{light}}{\to }R\cdot \;+\;H\cdot$$

The next stage is propagation, which is a series of reactions. Initially, a free radical reaction takes place with an oxygen molecule to form a peroxide radical, which then extracts a hydrogen atom from another polymer chain to form an unstable hydroperoxide group, which is divided into two new free peroxide or hydroxyl radicals. Because each initiating radical can generate 2 new free radicals, this process can be accelerated depending on how easy it is to remove the hydrogen from other polymeric chains, and how quickly the free radicals submit to termination by recombination and disproportionation. The reactions during the propagation stage are as follows [40,41]:$$R\cdot +\;O_{2}\to \mathrm{ROO}\cdot \;\mathrm{ROO}\cdot \;+\;\mathrm{RH}\to R\cdot \;+\;\mathrm{ROOH}\;\mathrm{ROOH}\to \mathrm{RO}\cdot \;+\;\cdot \mathrm{OH}\;\mathrm{RO}\cdot \;+\;\mathrm{RH}\to R\cdot \;+\;\mathrm{ROH}\;\mathrm{OH}\;+\mathrm{RH}\to R\cdot \;+H_{2}O$$

The final stage is termination, whereby two compounds with an unpaired electron form one inactive product. This can be a reaction between 2 peroxide radicals, alkyl radicals or 2 different radicals in the system [41]:$$R\cdot \;+\;R\cdot \;\to \;R-R\;2\mathrm{ROO}\cdot \;\to \;\mathrm{ROOR}+O_{2}\;R\cdot \;+\;\mathrm{ROO}\cdot \;\to \;\mathrm{ROOR}\;R\cdot \;+\;\mathrm{RO}\cdot \;\to \;\mathrm{ROR}\;\mathrm{HO}\cdot \;+\;\mathrm{ROO}\cdot \;\to \;\mathrm{ROH}+O_{2}$$

Focusing on degradable polymers, for example, polyolefins (polyethylene—PE, polypropylene—PP), two of the primary modes of degradation in industrial practice are thermal and photodegradation [42]. The initiation stage of polymer degradation via UV light is mainly associated with the presence of UV chromophores blended in with the polymer. Since saturated polyolefins do not themselves assimilate much UV light directly, the greatest impact of harmful UV is from absorption by chromophores in such compounds as pigments, flame retardants, catalyst residues, and any organic molecules that contain double bonds. As a result of the release of the absorbed part of the UV energy, the bond is cracked and free radicals are released, which initiate the degradation process. In the case of polypropylene, this process leads to chain scission, while for polyethylene, the cross-linking reaction is predominant [43]. Additionally, bonds of polyolefins themselves can be to some extent degraded by UV wavelength radiation—around 300 nm for PE and around 370 nm for PP [42]. An example of the loss of polyolefin properties due to the action of UV light is high density polyethylene (HDPE) in the form of a 1.5 mm wide plate, which can lose 80% of its strength after 2000 h of exposure to UV [43].

In the case of elastomers, they undergo various changes over time when exposed to ultraviolet light, heat, oxygen or ozone. Currently, there are a few standards that assess elastomers under elevated temperature or in chemical environment, such as ISO 188:2011 and ISO 1817:2015. When rubber is heated in the presence of air or oxygen, it loses strength, especially its tensile strength decreases. The determination of tensile strength change during material aging is not so simple because there are two opposing reactions that can take place simultaneously [44,45]. In order to explain the mechanical behaviour of crosslinked elastomer under exposure to various temperatures, Tobolsky et al. [46] offered a two-network theory. On the one hand, at higher temperatures softening can be observed, which is caused by the degradation of the molecular chains and crosslinks, and on the other hand, hardening can be observed as a result of additional cross linking. Depending on the type of vulcanization system, antidegradants used and type of filler, either softening or hardening reactions may prevail under any given aging conditions [44,45].

For example, crosslinking predominates in polybutadiene (BR) and its copolymers, such as nitrile-butadiene (NBR), styrene-butadiene-styrene (SBS), and in other diene rubbers with less active double bonds, whereas elastomers with electron donating side groups (–CH_3_) attached to a carbon atom vicinal to the double bonds are susceptible to chain scission. This includes natural rubber (NR), isobutylene isoprene rubber (IIR), polyisoprene (IR), and any other unsaturated elastomer with electron donating groups [47,48].

More detailed description of the general mechanism of thermal degradation will be presented in a future publication.

During the thermoxidation, mechanochemical or photochemical processes that occur under the influence of radiation, the radical chain process plays a key role. Counteracting these unfavorable processes can be achieved by improving stability, i.e., the material’s resistance to aging. There are various ways of modifying polymers by chemical and physical methods to increase resistance. The most common is modification with various additives such as stabilizers, antioxidants and UV absorbers [28,41,49].

## 3. Lifetime of Polymers

The durability of synthetic polymers is important for both manufacturers and users of plastic products, and above all for waste management. Unfortunately, not every type of polymer and plastic can be reprocessed by a recycling process. For this reason, the degradation processes of polymeric materials are constantly in the spotlight [50].

Lifetime (i.e., service life, shelf life, storage life) of polymeric materials at ambient or elevated temperature is a pivotal property that constitutes the acceptable lifetime (manufacturer’s warranty). After this time, the material reaches the threshold (usually 50% of the initial value) of the measured value at service temperature [51,52]. The storage time depends on the conditions under which the product is stored and on the type of elastomer and type of product (e.g., gaskets, tyres, etc.), as well as application (e.g., space, automotive, etc.) [15].

Rubbers belong to the group of materials characterized by long shelf life when stored in appropriate conditions, such as: temperature below 25 °C, lack of light, moisture, oxygen, ozone and chemicals, when the durability of the material can be for 3–25 years, depending on the polymer used [15]. The storage life of products made of different rubbers is given in Table 1 [53].

Accelerated ageing tests are carried out by simulation of natural conditions in laboratory equipment using intensification of factors influencing the polymer and accelerating the ageing process. The accurate prediction of the material lifetime under the conditions of use is very important in terms of safety (especially considering elastomeric construction materials), environment (replacement of traditional polymers with new biopolymers that are more eco-friendly) and in many others fields [18,30].

## 4. Accelerated Aging of Polymeric Materials

Estimating the life of polymeric materials through accelerated aging is an essential tool that provides a quantitative or qualitative comparison depending on the methodology applied [54,55,56].

The ageing of material under operating conditions may take a very long time before changes are visible, so degradation processes are accelerated [57,58,59]. It was assumed that, in natural conditions, the time needed to assess changes in material properties for soft plastics is about 3 years, while for hard plastics it is not less than 5 years. Therefore, application of accelerated aging tests in laboratory conditions significantly simplifies the process and its analysis. Based on the accelerated aging test, an approximate assessment of the aging resistance of the material is obtained [28].

Due to the importance of predicting the lifetime of polymeric materials in areas such as defense applications, nuclear reactor safety components and aircraft components, there is strong emphasis on developing increasingly better methods of accelerated ageing [17]. The durability of synthetic polymers is important both for manufacturers of plastic products and their users, and above all from the point of view of waste management. Unfortunately, not every type of polymer can be recycled by re-processing. Therefore, it is necessary to determine the lifetime of polymeric materials and to create materials capable of long-term use [50]. In order to research the accelerated ageing of the polymers, the conditions under which the product will be operated should be determined and several elevated temperature values are then selected for the ageing process. Usually, the rate of a chemical reaction increases with temperature. As a result of subjecting polymeric samples to increased temperature, it is possible to determine the relationship between temperature and the rate of the degradation reaction [15,51]. By extrapolation, the degree of degradation of the material after a certain time or the time required to achieve a certain degree of degradation can be estimated for a specific temperature [60].

Accelerated ageing tests are carried out under more aggressive conditions than potential operating conditions (higher concentration of oxygen and/or ozone in the atmosphere, higher temperature). Accelerated (short term) ageing methods are based on appropriate selection of the set of external factors (e.g., temperature, light, heat, moisture) and the set of measured properties (e.g., changes in mechanical or dielectric properties). Apart from the selection of external factors and their intensity, the speed of these changes is also important. Results obtained from accelerated aging give an approximate assessment of resistance to aging [56,60,61,62,63].

## 5. Methods for Predicting the Lifetime of Polymeric Materials

### 5.1. Analysis of Thermal Degradation Kinetics Using Thermogravimetric Analysis

The study of polymer degradation processes using thermal analysis methods has been the subject of interest of many researchers for several decades. This is also evidenced by the large number of research papers devoted to this issue [64,65,66,67,68,69]. Recently, Seifi et al. (2020) [70] presented the applications of thermal analysis techniques in research in the past decade (2010–2020). In past years, this was mainly related to the search for new thermally resistant polymers, nowadays, it concerns problems related to environmental protection, i.e., the search for polymeric materials with a specific lifetime [11,50,67,69,71].

Thermal analysis methods, including most often thermogravimetry (TG) and differential scanning calorimetry (DSC), are also used to record the course of polymer degradation [64,72,73]. These methods give the opportunity to determine the changes in the state of the test sample when the temperature changes under different measuring conditions [64]. The methods of thermal analysis are used to study phase changes and chemical reactions that occur when heating or cooling the substance. They also enable the determination of thermodynamic and kinetic parameters of the reaction [74,75]. As a result of subjecting polymer samples to elevated temperatures, it is possible to determine the relationship between temperature and degradation reaction rate. By extrapolation, it is possible to estimate, for a given temperature, the degree of material degradation after a specified time, or the time required to achieve a certain degree of degradation [76]. Many authors adopt different kinetic models and their corresponding conversion functions [77,78]. Methods using a combination of thermal analysis with other techniques, such as infrared analysis and mass spectroscopy of degradation products, have proved particularly useful [79,80].

An important aspect of the degradation process model is the kinetic description, which consists of determining the activation energy (E_a_), Arrhenius preexponential factor (A) and reaction order (n) [69,81,82,83]. These parameters are of theoretical and also practical importance. However, they must be supported by an understanding of the mechanism of the chemical reactions that occur in the process of polymer degradation. For non-isothermal tests, it should be assumed that the constant k is temperature dependent, e.g., according to the Arrhenius equation [64,84]. The differential equation representing the basic kinetic dependence of the degradation process has the form [78,85], Equation (2):(2)$$\frac{d\alpha }{\mathrm{dt}}=k\left(T\right)\cdot f\left(\alpha \right)$$
where k(T) is temperature-dependent reaction rate constant and f(α) is the model of reaction.

### 5.2. Arrhenius Model

The degree of degradation of polymers is measured by the deterioration of their properties [15,23,86]. Depending on the temperature, the rate of deterioration of these properties, i.e., the rate of degradation reaction, varies [86]. An increase in temperature usually results in an increase in the reaction rate, which is associated with an increase in the reaction rate constant (k) [87,88]. The relationship between the rate constant and temperature is described by the Arrhenius equation [78,89,90], Equation (3):(3)$$k=k\left(T\right)=A\cdot e^{\frac{-E_{a}}{\mathrm{RT}}}$$
where A is the frequency factor, E_a_ is the activation energy, R is the gas constant and T is the absolute temperature.

The above equation is usually presented in a logarithmic form [15,91], Equation (4):(4)$$\mathrm{lnk}\left(T\right)=-\frac{E_{a}}{\mathrm{RT}}+\mathrm{lnA}$$

The reaction rate is obtained from the change at any temperature in the particular property with exposure time at this temperature [92]. Due to different reaction rates k_i_ at different temperatures T_i_, the same property level x_a_ is reached after different reaction times t_i_ (Figure 3) [15,93], Equation (5):(5)$$F\left(x_{a}\right)=k_{i}\left(T_{i}\right)t_{i}$$
where F(x_a_) is the reaction state function.

By combining Equations (4) and (5), a logarithmic relationship is obtained [15,94], Equation (6):(6)$${\mathrm{lnt}}_{i}=\frac{E_{a}}{\mathrm{RT}}+B$$
where B is a constant.

Plotted against the inverse of temperature, the graph ln(t) forms a straight line and is known as the graphical form of the Arrhenius equation [95]. The lifetime of a polymeric material is read from the Arrhenius equation graph, which is extrapolated for a particular temperature and calculated from the following dependence [15], Equation (7):(7)$$e^{{\mathrm{lnt}}_{i}}=t_{i}$$

To eliminate the time dependence in Equation (2), which depends on temperature (T) and conversion rate (α), when heating at constant speed, the equation should be converted by dividing the differential equation by the heating rate [85], Equation (8):(8)$$\frac{d\alpha }{\mathrm{dT}}=\frac{A}{\beta }\cdot e^{\frac{-E_{a}}{\mathrm{RT}}}\cdot f\left(\alpha \right)$$
where β—$\frac{\mathrm{dT}}{\mathrm{dt}}$ is the heating rate.

By applying the time-dependent velocity equation and the linear transformation, kinetic parameters are obtained: activation energy (E) and response factor (A) [85], Equation (9):(9)$$\ln\left(\frac{\frac{d\alpha }{\mathrm{dT}}}{f\left(\alpha \right)}\right)=\ln\left(\frac{A}{\beta }\right)-\frac{E}{\mathrm{RT}}$$

Monitoring the ageing process with one exposure time is insufficient. It is, therefore, necessary to determine trend curves for material properties for several experimental conditions in order to extrapolate to the lifetime. Figure 4 presents an Arrhenius plot relating the time extrapolated as a function of inverse temperature.

Ln(t_i_) is proportional to E_a_/RT, E is the activation energy and R is the gas constant. This linear dependence enables prediction of the lifetime t_u_ extrapolated at T_u_ (temperature of use) [59].

The Arrhenius method assumes that the mechanism of degradation at elevated temperature is identical to that of degradation under the operating conditions. However, this assumption is not always true. The mechanism of this process at low temperature may be quite different, so that in such a situation it is not possible to extrapolate the value of material’s lifetime using a linear time–temperature relationship [88].

Lewandowski et al. (2016) [15] used the Arrhenius and time–temperature superposition methods to predict the lifetime of different elastomers (e.g., nitrile-butadiene rubber—NBR, chloroprene rubber—CR, ethylene propylene diene monomer—EPDM) and, in their opinion, predictions using the Arrhenius method can be subject to large error if the degradation is of a complex nature, e.g., physico-chemical, or the degradation mechanism changes depending on the temperature. Also, Xiong et al. (2013) [96] applied this method to lifetime prediction of NBR composite sheet in aviation kerosene by using nonlinear curve fitting of ATR-FTIR spectra. The lifetime of the NBR composite sheet in the aviation kerosene was 11,113 days at 20 °C, 6467 days at 25 °C, 3831 days at 30 °C, 2309 days at 35 °C and 1414 days at 40 °C. In the author’s opinion, the method based on the Arrhenius equation is valuable and the NBR composite sheet can still be safely used to store aviation kerosene after it has been used in aviation kerosene at room temperature for 8 years.

Madej-Kiełbik et al. (2019) [97] adopted the Arrhenius methodology for the accelerated aging of personal protectors for motorcyclists. The main constituents of the protectors were polyurethane and ethylene-vinyl acetate copolymer (EVA). They determined the endlife criterion (x_a_) using the kinetics of changes in the selected property/parameter of the material, at temperature intervals of 10 °C. Finally, they were able to calculate the real aging time of the material.

Koga et al. (2019) [98] analyzed the degradation behavior of PVC resin under high temperature conditions using Fourier transform infrared spectroscopy (FTIR), tensile testing and small punch (SP) testing. Based on the results from these tests, they compared the activation energies and estimated lifetime using the Arrhenius method. In this case, it turned out that SP testing is the most accurate and minimally destructive lifetime prediction method that can estimate early deterioration.

Wang et al. (2019) [99] proposed a lifetime prediction model of aging natural gas polyethylene (PE) pipeline with various internal pressures by thermal-oxidative aging (TOA and oxidative induction time (OIT) tests. The Arrhenius relationship was interpreted as a linear correlation between the OIT (logarithmic scale) at different test temperatures and the inverse of temperature (1/T). Lifetime prediction over the range 0–0.4 MPa at 20 °C proved to be in excess of the 50 years’ lifetime requirement. The authors confirmed that this method is very suitable for pressured urban gas PE pipes, and also very suitable for other plastic pipes in similar environments.

Based on these examples, it can be said that the Arrhenius model is useful for many different polymeric materials, e.g., NBR, EPDM, EVA, PVC, PE, etc. The results that are obtained with this method are valuable but only in the case when degradation is not limited by diffusion. The approach may be used for lifetime prediction within a temperature range where the degradation mechanism remains the same. Most authors choose this way of lifetime prediction because the method is reliable and time saving, and can calculate change of aging performance at any time.

### 5.3. Time–Temperature Superposition

The principle of time–temperature superposition (TTSP) is one of the bases of accelerated experimental methodology for polymeric materials [100]. The accuracy of predicting the lifetime of polymeric materials and activation energy can be improved by using the time–temperature superposition method [15].

This method is determined from an empirical assumption that the influence of temperature and time on the characteristics are equivalent. For a short-period of accelerated aging at higher temperature, the changes of characteristic variable are the same as those measured for longer times but at normal (lower) temperatures. The TTSP consists of determining the a_T_ factor (shift factor), which results in a single curve describing the correlation between the tested material property and temperature T or time t [91], Equation (10):(10)$${\mathrm{lna}}_{T}=\frac{E_{a}}{R}\left(\frac{1}{T_{0}}-\frac{1}{T}\right)$$
where a_T_ is a shift factor, T is a temperature and T_0_ is a reference temperature (usually the lowest value of the ageing temperature).

A simple scheme for this method is given in Figure 5.

For polymeric materials, it is possible to determine temperature functions that enable displacing respective isothermal segments of the selected response function along the logarithmic time scale and create a master curve, which is recorded at a reference temperature (T_0_) [100,101,102].

By creating a straight line in the logarithmic diagram of empirically determined values of the shift coefficient in relation to the absolute temperature, the shift coefficient can be calculated for each desired temperature. Once the displacement coefficients are determined, they are used to extrapolate the changes in behaviour at ambient temperature [91]. Such a curve is called a complex curve and, unlike the Arrhenius method, it uses all experimental points for all ageing temperature values. From the complex curve it is possible to determine the time needed to obtain a threshold value at T_0_ [103].

The lifetime of a polymeric material at operating temperature can be determined by extrapolating the log a_T_ value for operating temperature (a_Te_) [15], Equation (11):(11)$$t_{e}=\frac{a_{T0}}{a_{\mathrm{Te}}}\cdot t_{0}$$
where a_T0_ is a shift factor for the reference temperature (a_Te_ is an extrapolated shift factor for operating temperature (determined from the dependence of the logarithm (a_T_) on the temperature) and t_0_ is the lifetime of the material at the reference temperature.

Nakada et al. (2011) [104] used the time–temperature superposition principle to predict the long-term viscoelastic behavior of amorphous resin. The master curves of creep compliances could be constructed from measured data by shifting the a_T_ factors vertically as well as horizontally. Therefore, the long-term viscoelastic behavior at a temperature below T_g_ can be predicted accurately based on the short-term viscoelastic behavior at elevated temperatures using the TTSP with vertical shift as well as horizontal shift. The creep compliance was also tested by Fukushima et al. (2009) [105] using the TTSP method for long-term lifetime prediction of polymer composites. In the authors’ opinion, it can be considered that the time–temperature shift factor is obtained accurately and easily from dynamic viscoelastic tests.

This has been confirmed by Krauklis et al. (2019) [101]. They applied the TTSP method to predict creep of a plasticized epoxy, which is a matrix for fiber reinforced polymer (FRP) composite materials. This methodology has turned out to be useful to predict the long-term viscoelastic behavior of plasticized polymers at temperatures below the T_g_ temperatures based on short-term creep experimental data.

Yin et al. (2019) [106] investigated the aging behavior of PMMA in a liquid scintillator at different temperatures under static tensile stress with dynamic mechanical analysis (DMA) and differential scanning calorimetry (DSC). Then, the service life of the PMMA was predicted based on the time–temperature superposition approach. The tensile strength of PMMA at different aging temperatures showed a downward trend with time of aging. For tests at 30 °C, 40 °C, 50 °C, linear behavior of the plot of the shift factors versus the inverse aging temperature was observed, indicating that the aging mechanism at these temperatures was consistent with the Arrhenius rule. The result at 55 °C deviated significantly from the straight line because the Arrhenius equation is valid only when the assumption of constant acceleration for chosen temperatures is fulfilled. However, when the material deviates from this assumption, the Arrhenius equation ceases to be useful, which could be observed in the case of the temperature of 55 °C. According to the Arrhenius model and extrapolation, the lifetime of PMMA was determined to be 25 years at 20 °C. A discussion about why the PMMA at 55 °C aged differently and what it means would be welcomed, although this is not needed for predicting lifetime.

Most papers that apply the time–temperature superposition principle concentrated on prediction of the long-term viscoelastic behavior of material at a temperature below T_g_. It should be emphasised that this method is valid only in accordance with the assumption of constant acceleration. In other cases, it can be subject to measurement error and the value of temperature will deviate from the straight line.

### 5.4. Williams–Landel–Ferry (WLF) Model

The Williams–Landel–Ferry (WLF) method is also used to predict the lifetime of polymeric materials based on any physical property [107]. In the case of complicated dependence of a given material property on time or when the degradation process is limited by diffusion, better results can be obtained by using the WLF method [15,108].

The WLF equation is an essential tool to predict temperature induced physical/chemical changes as a function of processing and storage conditions. This equation allows for the estimation of the shift factor (a_T_) for temperatures other than those for which the polymeric material was tested. Compared to the Arrhenius equation that describes the behaviour above the glass transition temperature (T_g_), the WLF equation is applicable close to this temperature [109].

The WLF model uses a time–temperature superposition without any additional assumptions regarding the dependence of material properties on time at any temperature. In this method, the a_T_ factor is described by the formula [110,111,112], Equation (12):(12)$${\log\;a}_{T}=\frac{C_{1}\left(T-T_{0}\right)}{C_{2}+\left(T-T_{0}\right)}$$
where a_T_ is a reduced variables shift factor, C_1_ and C_2_ are the experimental constants, which depend on temperature and the material being tested, and T_0_ is a reference temperature (usually T_g_
*≤* T_0_
*≤* T_g_ + 100).

The WLF method can be used when the dependence of a given property on time at different temperature values is similar. The curves corresponding to the different temperature values are shifted parallel to each other by the shift factor a_T_ [112].

By comparing Equations (10) and (11) and converting the logarithms from natural to decimal, the following values of C_1_ and C_2_ are obtained [113], Equations (13) and (14):(13)$$C_{1}=-\frac{1}{2.303}\cdot \frac{E_{a}}{{\mathrm{RT}}_{0}}$$
(14)$$C_{2}=T_{0}$$

This equation has been successfully applied to polymeric materials using the “universal values”. However, the literature has shown that these constants can be different and should be measured experimentally [114].

Hu et al. (2013) [113] applied the WLF and Arrhenius equations to research on temperature and frequency dependent rheological behaviour of carbon black filled natural rubber. They confirmed that the temperature dependence of the shift factor is modelled well by both the Williams–Landel–Ferry equation and the Arrhenius equation.

Ljubic et al. (2014) [115] considered the possibilities of using time–temperature superposition with the WLF equation. This method is well established for bulk linear and homopolymers but for more complex polymeric materials (individual, blend or composite) with different structures and morphologies, it can sometimes be a challenge. The combination of TTSP with the WLF equation can be successfully applied to crosslinked polymers (polyurethanes and epoxy), polyolefins for biomedical application, Kevlar 49, polymer blends, biopolymers and polymer composites. In these cases, the dynamic mechanical and viscoelastic properties were tested, and modeling of the properties was performed by using TTSP and the WLF equation. In the authors’ opinion, this combination of TTSP with WLF has versatile applications and can be a useful tool in the study of a broad range of polymeric materials, their properties and lifetime prediction for final products.

Chang et al. (2013) [116] used the TTSP method with the WLF equation to apply to a series of short-term creep tests of a wood–plastic composite (WPC). The success of the method implied that the WPC product studied is a thermo-rheologically simple material and only horizontal shifting is needed for the time–temperature superposition. Additionally, the temperature shift factor used to construct the master curve was fitted well by the WLF equation. Dan-asabe B. (2016) [117] used a long term TTSP (time–temperature superposition) performance prediction with WLF assumption to characterization of a banana (stem) particulate reinforced PVC composite as a piping material. The composite turned out to be rheologically simple as it satisfied the WLF condition. Application of long-term prediction showed that the composites have better long-term performance than PVC pipe over a period of 126 years of use.

As has been observed, the WLF equation is always used in combination with the TTSP principle. By applying this method, it is possible to obtain better lifetime prediction results than with the Arrhenius method when the time dependence of a property is complicated or when degradation is limited by diffusion.

### 5.5. Isoconversional Methods

Isoconversional methods are one of the more reliable kinetic methods for processing thermoanalytical data [118]. They are based on the isoconversional principle, with the main presumption of temperature independence of the pre-exponential factor and the activation energy. However, both the pre-exponential factor and the activation energy are still interrelated conversion functions [119]. These methods have their roots in the Ozawa–Flynn–Wall [120,121] and Friedman [122] methods developed in the 1960s.

The accuracy of the method has been improved by the Kissinger–Akahira–Sunose (KAS) [123] approach to calculating kinetic parameters. Starink [124] and then Gao et al. [125] proposed subsequent modifications to the calculation procedure and coefficients in the KAS equation to predict more accurate values of the pre-exposure factor and the activation energy. Vyazovkin has continuously developed an alternative calculation method of activation energy [126,127], giving access to the well-known advanced isoconversion method (AICM), now widely accepted as one of the most precise methods of estimating the activation energy from TGA experiments [128]. All of the abovementioned methods, except the Friedman method, are integral, i.e., the parameters are calculated on the basis of integral analysis of the measured TG signals [119].

The main benefits of isoconversional methods are that they allow the evaluation of the effective activation energy, E_a_, without presuming any specific form of the reaction model, f(α) or g(α)*,* and that a change in the variation of E_a_, may generally involve a change in the reaction mechanism or the degree of limitation of the overall rate of reaction, as measured by thermoanalytical techniques [118].

Isoconversion methods require a series of experiments with different temperature programs and obtaining the effective value of activation energy E_a_ as a function of conversion degree α. A significant variability of E_a_ with α means that the process is kinetically complex, and the activation energy dependences assessed by an isoconversional method enable significant mechanistic and kinetic analyses and the understanding of multi-step processes, as well as reliable kinetic prognoses. Model-free isoconversional methods are a powerful tool to obtain information about the complexity of the reaction by determining E_a_ [118,129,130].

#### 5.5.1. Friedman’s Method

The determination of activation energy without the need to adopt a specific kinetic model is possible using this method. The E_a_ value is based on thermal measurements at various heating rates. The Friedman’s isoconversion method is a widely used differential method which, unlike conventional integral isoconversional methods, provides accurate values of the activation energy [119,131]. The modification of the general reaction rate equation results in the following expression [118], Equation (15):(15)$$\ln\frac{d\alpha }{\mathrm{dt}}=\ln\left[\mathrm{Af}\left(\alpha \right)\right]-\frac{E_{a}}{\mathrm{RT}}$$
where α is a specific degree of degradation.

The term [Af(α)] constitutes the product between the mathematical function f(α) and the pre-exponential factor A that characterizes the reaction mechanism. After evaluating E_a_ and [Af(α)], the response rate (dα/dt) for each α value can be calculated [118].

By using the following linear regression, the activation energy at each level of isoconversion can be determined [132], Equation (16):(16)$$\ln\frac{d\alpha }{\mathrm{dt}}=f\frac{1}{T_{\mathrm{jk}}}$$
where T_jk_ are the temperatures at which the degree of conversion is achieved (α_k_) at the heating rate (β_j_).

Das et al. (2017) [133] made a comparison of different kinetic models in order to research the thermal decomposition behaviour of high and low-density polyethylene (LDPE and HDPE), polypropylene (PP) and poly(lactic acid) (PLA). One of them was the Friedman method. According to the opinion of these authors, it is difficult to choose one model that can extract the correct kinetic parameters from the complex reactions occurring, and it is better to use several models and make a comparison. Cui et al. (2014) [134] used the Friedman method to study the thermal degradation kinetics of photonically cured electrically conductive adhesives. To demonstrate the applicability of the Friedman method, the results calculated from this kinetic method were compared with those from experiments and they turned out to be in good agreement. In the authors’ opinion, with the Friedman method, a comprehensive and in-depth understanding can be obtained of the thermal degradation kinetics of photonically cured electrically conductive adhesives.

Mittal et al. (2020) [135] studied the thermal decomposition kinetics and properties of grafted barley husk reinforced PVA/starch composite films for packaging applications, based on TGA measurements. They determined the activation energy for composite films by using Friedman (FR), Flynn–Wall–Ozawa (FWO), Kissinger–Akahira–Sunose (KAS), and modified Coasts Redfern (CR) methods. It was observed that the activation energy obtained using these all methods showed the same trend for various conversion factors (0.1−0.9) for all the films, which indicated the reliability of the values of activation energy obtained.

#### 5.5.2. Ozawa–Flynn–Wall (OFW) Method

The Ozawa–Flynn–Wall (OFW) method is an integral isoconversional technique, in which the activation energy is related to the heating rate and temperature at a constant conversion rate [136]. The main advantage of this method is that it does not require any assumptions about the form of the kinetic equation except a temperature dependence of the Arrhenius type. This method is a model-free method that measures temperatures corresponding to constant values of α from experiments at various heating rates (β), and plotting ln(α) in relation to 1/T. The slopes of such graphs give −E_a_/R [137]. In this method, by using Doyle’s approximation of the temperature integral, the following equation is obtained [130,136], Equation (17):(17)$$\ln\beta =\ln\frac{{\mathrm{AE}}_{a}}{\mathrm{Rg}\left(\alpha \right)}-5.331-1.052\frac{E_{a}}{{\mathrm{RT}}_{p}}$$
where β is a heating rate, E_a_ is the activation energy, R is the gas constant and T_p_ is the peak temperature.

According to the above equation, from the experimental thermogravimetric curves, which are recorded for several heating rates, for α = const, the linear regression of the relation ln(β) = f(1/T) is obtained and its slope can be used to determine the activation energy [136].

The OFW method is potentially suitable for use in systems with multiple reactions such that the activation energy changes over time. However, it is predicted that this method will fail if reactions of very different kinds occur simultaneously. Competitive reactions that have various products also make this method inapplicable. Moreover, the OFW method is less accurate than the Friedman’s method. It has been shown that, if E_a_ depends on the degree of conversion, its values obtained by integral and isoconversional methods are different [137].

Ali et al. (2019) [138] calculated the activation energy E_a_ by applying Coats–Redfern, Ozawa–Flynn–Wall, Kissinger–Akahira–Sunose and Friedman models in order to research the thermo-catalytic decomposition of polystyrene waste. From the results obtained, it was observed that activation energy investigated using the OFW is the lowest, therefore this model is most suitable for explaining catalytic decomposition of expanded waste polystyrene.

Vassiliou et al. (2010) [139] used the OFW method to study the thermal degradation kinetics of organically modified PET with montmorillonite and fumed silica nanoparticles. It enabled them to determine a kinetic parameter in terms of the apparent activation energy.

Benhacine et al. (2014) [140] applied the Flynn–Wall–Ozawa method to determine the activation energy E_a_ of the degradation process of isotactic polypropylene/Algerian bentonite nanocomposites prepared via melt blending. Determination of activation energy was based on weight loss versus temperature data obtained at several heating rates, and this was necessary in order to deeply analyze the effect of incorporation of pure bentonite or organically modified bentonite into iPP matrix on the degradation mechanism of iPP.

#### 5.5.3. Ozawa–Flynn–Wall Corrected Method by N. Sbirrazzuoli et al.

N. Sbirrazzuoli et al. [141] proposed another way to obtain corrected values of activation energy by applying a numerical integration of p(x). The first part of this method involves calculating Equation (18) in order to obtain an approximate value of activation energy for a given conversion (Ozawa–Flynn–Wall method). Next, the average temperature ($\bar{T_{\alpha }}$) for different heating rates is assessed and p(x) is extrapolated by numerical integration of the following equation, Equation (18):(18)$$g\left(\alpha \right)=\frac{\mathrm{AE}}{R\beta }\left[\frac{\exp\left(-x\right)}{x}-\int_{x}^{\infty }\left(\frac{\exp\left(-x\right)}{x}\right)\mathrm{dx}\right]=\frac{\mathrm{AE}}{R\beta }p\left(x\right)$$
where x = E/RT is minimized activation energy at the temperature T. This equation assumes that the value of E is constant.

Finally, ln p(x) values are matched by a first order polynomial in the range $x=E_{\alpha }\;\left(1\pm 0,2\right)/\left(R\bar{T_{\alpha }}\right).$ [141].

#### 5.5.4. Kissinger–Akahira–Sunose (KAS) Method

The KAS method is considered to be one of the best isoconversional methods because of the Coats–Redfern approximation of the temperature integral. It does not require knowledge of the exact thermal degradation mechanism [136,142]. Using this method, the activation energy is determined from the following equation [143], Equation (19):(19)$$\ln\left(\frac{\beta }{T_{p}{}^{2}}\right)=-\frac{E_{a}}{{\mathrm{RT}}_{p}}+\mathrm{const}$$
where T_p_ is a peak temperature.

Therefore, for α = const. the plot of ln(β/T_p_^2^) against (1/T_p_) obtained from thermogravimetric curves, recorded for different heating rates, is a straight line. The slope and intercept of this line can yield the pre-exponential factor and activation energy, respectively [136,143].

Wang et al. (2005) [144] described the possibility of using the KAS algorithm for modeling the cure kinetics of commercial phenol-formaldehyde (PF) resins. Additionally, this algorithm could also be used to predict the isothermal cure of PF resins from dynamic tests. It has also been confirmed by Gabilondo et al. (2007) [145] that the d KAS method seems useful for the dynamic cure prediction of that type of thermoset.

Additionally, Sun et al. (2014) [146] applied the Kissinger–Akahira–Sunose (KAS) method to determine activation energy (E_a_), and investigated it as the change of conversion (α) in order to describe the curing behavior of epoxy resins by using differential scanning calorimetry.

#### 5.5.5. Advanced Isoconversional Method by S. Vyazovkin

The advanced isoconversional method developed by Vyazovkin S. [126] is a non-linear integral method and is free of the approximations applied in OFW and KAS approaches. It is based on a direct numeric integration of the following equation [141], Equation (20):(20)$$g\left(\alpha \right)=\frac{A}{\beta }\int_{0}^{T}\exp\left(-E/\mathrm{RT}\right)\mathrm{dT}$$
where β = dT/dt is a heating rate.

For a set of n experiments performed under various temperature programs T_i_(t), the activation energy is assessed at any specific value of α by finding the E_a_ value which minimises the function, Equation (21) [147]:(21)$$\phi \left(E_{\alpha }\right)=\overset{n}{\underset{i=1}{\sum }}\overset{n}{\underset{j\neq i}{\sum }}\frac{J\left[E_{\alpha },T_{i}\left(t_{\alpha }\right)\right]}{J\left[E_{\alpha },T_{j}\left(t_{\alpha }\right)\right]}$$

Recently, this method has been modified by integration, which is performed over small time segments that enables elimination of the errors related to integral methods when activation energy varies significantly with α, Equation (22) [147]:(22)$$J\left[E_{\alpha },T_{i}\left(t_{\alpha }\right)\right]=\overset{t_{\alpha }}{\underset{t_{\alpha }-\Delta \alpha }{\int }}\exp\left[\frac{-E_{\alpha }}{{\mathrm{RT}}_{i}\left(t\right)}\right]\mathrm{dt}$$

In Equation (22), α varies from ∆α to (1 − ∆α) with a step ∆α = m^−1^, where m is the number of intervals that are chosen for analysis [141].

The first experimental deployments of the isoconversional predictive procedure were made by Vyazovkin [148] and Vyazovkin and Sbirrazzuoli [149] concerning the application of nonisothermal data to predict epoxy curing kinetics under isothermal conditions. For both methods, the isoconversional predictions agreed well with the real measurements and were better that those obtained by other methods.

Dunne et al. (2000) [150] predicted isothermal and nonisothermal cure kinetics of an epoxy-based photo-dielectric dry film (ViaLux™ 81). They reported perfect agreement with actual nonisothermal measurements. However, the predictions for isothermal conditions deviated markedly from the actual data. The reason for this was that they were done to a temperature below the limiting glass transition temperature.

Li et al. (2001) [151] described several examples of successful kinetic predictions obtained by applying the isoconversional methodology to the curing process of polyurethane. He et al. (2005) [152] applied isoconversional predictions to demonstrate the accelerating effect of moisture in wood on curing of polymeric diphenylmethane diisocyanate.

Polli et al. (2005) [153] used the isoconversional analysis in order to predict the kinetics of thermal degradation of branched and linear polycarbonates in a wide temperature range. Vyazovkin et al. (2004) [154] certified that the isoconversional predictions of degradation of polymeric materials to the flash ignition temperature can be very usable in assessing potential fire resistance of polystyrene (PS) and PS–clay nanocomposite.

Over recent years, the use of isoconversional kinetic analysis has provided new opportunities in traditional areas of application such as polymer degradation or curing, and also efficiently delving into areas such as glass transition. The effectiveness of these methods comes from their ability to handle the complexity of different processes. The resulting activation energies can be used to perform reliable kinetic predictions, to obtain information about complex mechanisms, and also to access inner kinetic parameters.

## 6. Conclusions

Degradation of polymeric materials can be caused by many different factors: both chemical (working environment) and physical (temperature, radiation) and mechanical (stress). The mechanism of degradation can, therefore, be very complicated. There are many different kinetic models that are used to predict the lifetime of these materials. Those described in this paper are those used most often.

Predictions are among the most important practical features of kinetic analysis. They are widely used to evaluate the kinetic behavior of polymeric materials beyond the temperature regions of experimental measurements.

For the Arrhenius model, it assumes that the mechanism of degradation at elevated temperatures is identical to the mechanism of degradation under operating conditions. However, this assumption is not always true. For this reason, a time–temperature superposition is used, which increases the accuracy of predicting the lifetime of polymeric materials and activation energy. It should be remembered that degradable polymers degrade during thermal processing, which can of course affect the lifetime predictions. The lifetime of polymers can also be determined by the Williams–Landel–Ferry method (WLF), in which there are no assumptions concerning the dependence of the tested property on temperature and, in the case of a degradation process that is limited by diffusion, better results can be obtained by using this method.

There are also isoconversional methods, which are considered to be one of the more reliable kinetic methods of processing thermoanalytical data, and their main benefit is that they allow the evaluation of the effective activation energy, E_a_, without presuming any specific form of the reaction model. The activation energy calculated from different isoconversional methods used mainly for prediction of lifetime is a feature to follow physico-chemical processes occurring inside the material. In fact, that the ageing process can result in scission, crystallization, oxidation, thermal decomposition, etc., and those factors are closely related. Moreover, in the great majority of publications, activation energies are determined using techniques like DTA and DSC, where endo- and exothermal transformations can be observed. Those changes can also be found during the ageing process. Therefore, activation energy, which can also be thought of as the magnitude of the potential barrier of molecules at the surface of the material, can bring necessary information on how fast the ageing process will occur. For instance, thermal stability can be estimated as the time to reach a certain extent of conversion at a given temperature. Kinetic predictions of this type can be easily achieved by using the activation energy dependence measured by an isoconversional method.

Scientists are constantly working to develop new methods or refine existing ones because of the great interest and need for such research on accelerated ageing and prediction of the lifetime of polymeric materials that are used in almost every area of our lives.

## Figures and Tables

**Figure 1 materials-13-04507-f001:**
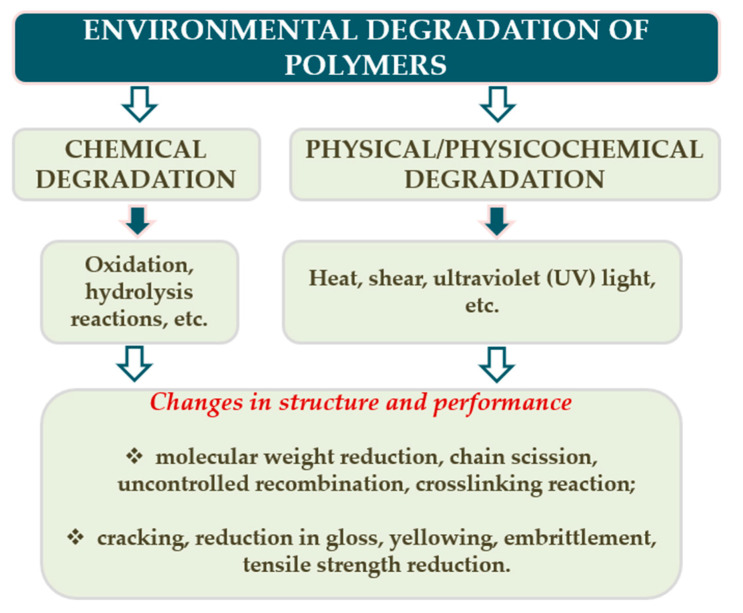
Changes in polymers caused by the chemical (oxidation, hydrolysis) and physicochemical (heat, shear, UV light) degradation factors.

**Figure 2 materials-13-04507-f002:**
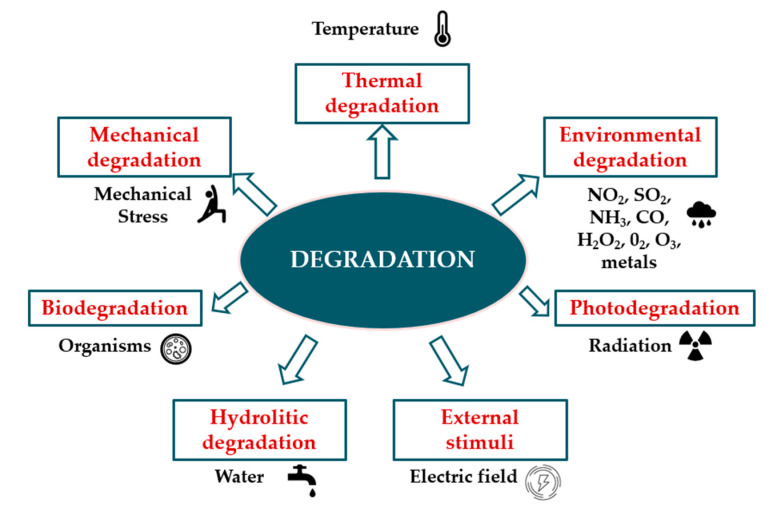
Chemical, physical and biological agents causing various types of degradation.

**Figure 3 materials-13-04507-f003:**
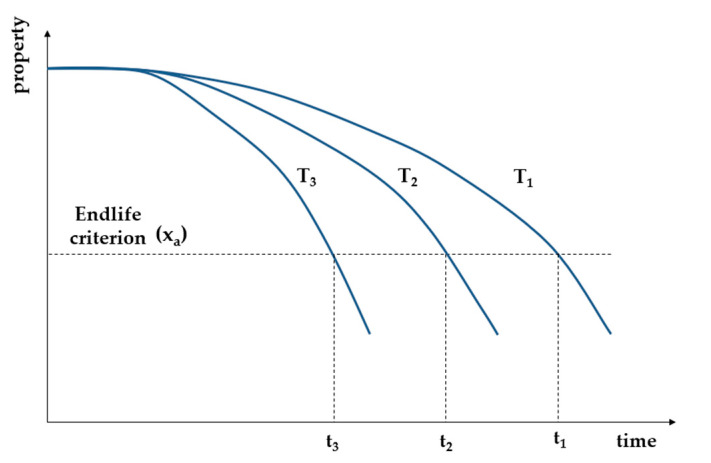
Change in property with time at three different temperatures T_1_ < T_2_ < T_3_ [59].

**Figure 4 materials-13-04507-f004:**
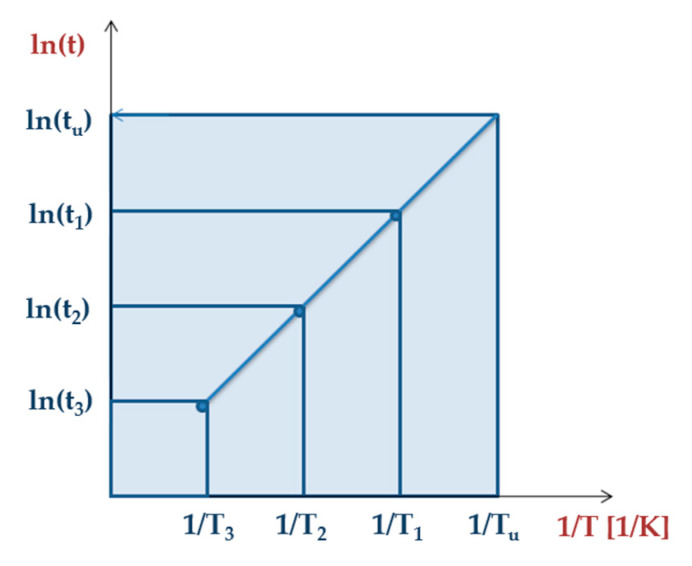
Arrhenius diagram showing time extrapolation as a function of inverse temperature [59].

**Figure 5 materials-13-04507-f005:**
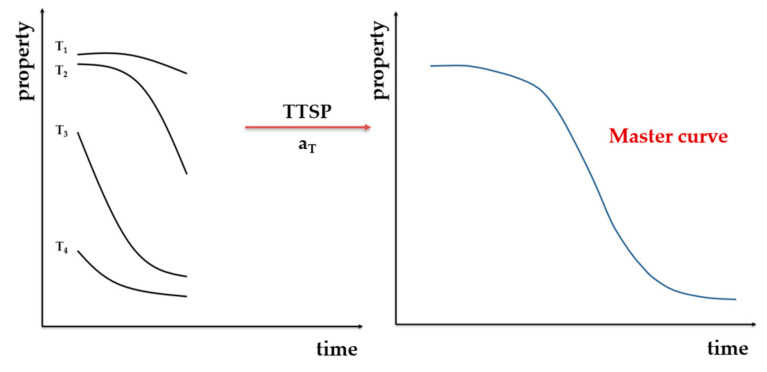
Schematic illustration of creating a master curve through the time–temperature superposition (TTSP) method [92].

**Table 1 materials-13-04507-t001:** Shelf life of products made out of different types of rubber [51].

Rubber	Abbreviation	Recommended Storage Life Without Inspection (Years)	Storage Life Extension After Visual Inspection (Years)
Natural Rubber	NR	5	2
Butadiene-styrene	SBR	5	2
Nitrile	N	7	3
Nitrile-butadiene	HNBR	7	3
Acrylic	ACM	7	3
Chloroprene	C	7	3
Ethylene-Propylene	E	10	5
Viton™/FKM	V	10	5
Kalrez™/FFKM	KLZ	10	5
Silicone	S	10	5

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
