# Peer review of "Lifetime Prediction Methods for Degradable Polymeric Materials—A Short Review"

_materials, 2020, doi:10.3390/ma13204507_

Round 1
Reviewer 1 Report
The authors have satisfactorily addressed concerns raised in the previous reviews. The revised manuscript may be considered for publication.
Author Response
Institute of Polymer and Dye Technology
Technical University of Lodz
90-924 Lodz, ul Stefanowskiego 12/16, Poland
Tel.: +48 42 631 32 23, Fax: +48 42 636 25 43
October 2nd, 2020
Materials— Open Access Journal
Dear Editor,
We have carefully considered the Reviewers' comments and the manuscript was revised exactly according to them. We improved the following parts of our paper entitled “Lifetime Prediction Methods for Degradable Polymeric Materials – A Short Review” with a request to consider it for publication in " Materials”:
After first review reports:
Reviewer 1:
- suggested that the statements such as "it is currently difficult to find a field in which plastics are not being used" is not necessary. Similarly, a discussion of thermoplastics and thermosets is also not required. We removed these parts from our manuscript.
- second comment was related to different equations explanation used in the manuscript. We explained step by step all doubts and corrected the mistakes. Also we improved the following sentence: The TTSP is based on the perception that the short-term and long-term behavior of viscoelastic materials is similar at higher temperatures." I am not sure what this statement means. Now, is: This method is determined from an empirical assumption that the influence of temperature and time on the characteristics are equipollent. For a short-period accelerated aging test at higher temperature the change of behavior of characteristic variable is the same to those measured for longer period but at normal condition.
- third comment was related to the section on isoconversional methods (5.5). According to the reviewer, this part was merely a summary of very well-known methods, without providing any novel insights. We added the part related to Ozawa–Flynn–Wall corrected method by N. Sbirrazzuoli et al. and we also described an advanced isoconversional method by S. Vyazovkin, which is an alternative calculation method of activation energy. Additionally, we presented the recently change in Vyazovkin method that has been modified by integration, which is performed over small time segments that enables elimination of the errors related to integral methods when activation energy varies significantly with α.
Reviewer 2:
- suggested to give more information regarding the significance of lifetime prediction of materials. In the section 4. Accelerated aging of polymeric materials it has been improved. But also we decided to change the title of this paper: Lifetime Prediction Methods for Degradable Polymeric Materials – A Short Review.
- suggested to specify to which type of polymers these considerations are addressed and under which conditions. It has been improved. We highlighted that this paper is concentrated on degradable polymers and methods of lifetime prediction using accelerated thermal ageing tests and extrapolation of data from induced thermal degradation.
- suggested to support the description of all the models discussed with examples of polymers to which they have been practically applied to and graphs to show the reader how to operate. In each section related to kinetic models, the examples of application in literature have been presented. We added 2 more graphs in section 5.2 and 5.3 but for other methods the specific data are necessary. We explained that our next aim is to do the research (accelerated aging of various polymers) and predict the lifetime using the different kinetic models. We are going to create the correlation between the methods described in this paper.
- last comment was related to a more appropriate refer 94 and we also improved it.
Reviewer 3:
- suggested to cite the literature examples in further detail. We improved the style of citing the literature.
- suggested to provide more advanced level details in section 2.3 which focuses on the basic principles of thermal degradation of polymer degradation. In this section we added more information about the UV degradation of polyolefins that are the example of degradable polymers.
- suggested to include more figures and improve the quality (font size, resolution and clarity) of existing figures. We included 2 more graphs in section 5.2 and 5.3 but for other methods the specific data are necessary. The quality of existing figures has been improved.
After second review reports:
Reviewer 1:
- suggested that it would be more helpful if the review article focused on how the methods have been applied by researchers and what their key conclusions were. The manuscript has been improved as recommended. In the section 2.3 we added more specific information related to thermoplastics and elastomers. In the section 5 (Methods for predicting the lifetime of polymeric materials) we included a lot of examples of how these models were used by other authors and what their conclusions were. Now, a potential reader can find the information to which type of polymers the specific model was used and what was observed. Additionally, the conclusions of the manuscript have been corrected.
Reviewers 2 and 3were satisfied with the enhancement of the manuscript.
After third review reports:
Reviewers 2 and 3 encouraged the publication of our manuscript.
Reviewer 1:
- wrote that the work is actually fine but does not see any possibility of publishing this in the present journal. The reasons were as follows: there are too many general things like Arrhenius equation discussed in detail, there are no concrete examples. According to the reviewer, this extensive topic is not the subject of a brief review article and suggested to pick out some examples, discuss them in detail and resubmit.
Each time we received a comment from this reviewer with different expectations without meaningful details. Based on the literature examples that were added after second review, we strongly believe that this gives significant value for the current paper, mainly take into account that the references used as examples of applications of kinetic models were carefully selected and come from recent years.
For correspondence please use the following information:
corresponding author: Anna Masek
Institute of Polymer and Dye Technology
Technical University of Lodz
90-924 Lodz, ul Stefanowskiego 12/16, Poland
Tel.: +48 42 631 32 93
Fax: +48 42 636 25 43
e-mail: anna.masek@p.lodz.pl
Yours sincerely,
PhD, Dsc Anna Masek
Reviewer 2 Report
Plota and Masek summarized recent investigations of lifetime of polymer materials. This topic is very important for not only basic science but also development of novel functional materials. In this short review, the authors described in detail various methods of lifetime predictions of the polymers. Therefore, I think the review deserves publication in the journal, however, I feel that the manuscript may be properly revised by taking a following comment.
Comment: In Figure 2, the authors summarized several types of degradation of polymers. I think “external stimuli” such as electric field affect for deterioration of polymer molecules/materials. The authors can add the factor in the figure.
Author Response

(The authors gave the same response as above.)

Reviewer 3 Report
Dear authors,
I have looked through your work carefully. Basically, the work is actually fine. However, I do not see any possibility of publishing this in the present journal. The reasons are as follows: there are too many general things like arrhenius equation discussed in detail, which can also be found in the textbook. There are no concrete examples. In fact the situation in practice is much more complex. The degradation depends on the shape, the internal tension of the system, the stabilizers that are used and of course fillers and pigments play a role. This extensive topic is not the subject of a brief review article. I suggest that you pick out some examples, discuss them in detail and resubmit. I'm sorry I couldn't make a better decision.
Author Response

(The authors gave the same response as above.)

Round 2
Reviewer 3 Report
Dear authors, of course I have not made any clear recommendations. But, that's not that easy either. The degradation of polymer materials depends on as many factors as additives, fillers, pigments, stabilizers. I've worked in industry too long to predict things on a clear theoretical basis. It is always necessary to carry out tests (quick tests). Nevertheless, I no longer stand in the way of this work. Good luck with that and all the best!!
This manuscript is a resubmission of an earlier submission. The following is a list of the peer review reports and author responses from that submission.
Round 1
Reviewer 1 Report
1) The manuscript is a review of the lifetime prediction of polymeric materials. This is an important topic in the field of polymer science and the review is comprehensive in terms of the number of publications covered. The article, however, is found to be lacking in technical rigor. Of the 20 pages of the article, the first four pages are elementary polymer science. For example, even a moderately advanced reader of the article would know about the significance of polymer in the present-day life, and statements such as "it is currently difficult to find a field in which plastics are not being used" is not necessary. Similarly, a discussion of thermoplastics and thermosets is also not required.
2) There are many statements that are unclear. For example:
2.1) Page 7: "Due to different constants of the reaction rates ki at different temperatures Ti, the same reaction threshold xa is reached after different reaction times ti." Equation (4) is cited. But there is no variable called "xa" in this equation. Furthermore, what is F(x) in Equation 4? What is the basis of the assumption that it is linearly proportional to time, ti?
Assuming that Equation 4 is correct, it is not clear how the combination of Equation 3 and Equation 4 results in Equation 5. How is the variable "B," called the reaction constant, related to the parameters of equations 3 and 4.
2.2) Similarly, other equations in this section (5.2 Arrhenius model) are completely unclear. First, there is a mistake in Equation 8 (dα/dt in Equation 8 should actually be dα/dT). Next, Figure 3 is not properly explained, and Equation 9 and a statement associated with it ("the correct choice of kinetic parameters can be achieved with many linear regressions") is not technically rigorous, even though it may have been copied verbatim from reference # 89.
2.3) In section 5.3: "The TTSP is based on the perception that the short-term and long-term behavior of viscoelastic materials is similar at higher temperatures." I am not sure what this statement means.
2.4) The concept of time-temperature superposition has not been properly explained. In Equation 11, why is the value of the shift factor equal to 1 at the reference temperature?
2.5) One page 10, the authors have stated that the Arrhenius equation describes the behavior below the glass transition temperature, and the WLF equation is applicable above the glass transition temperature. However, WLF equation is generally found to be applicable close to the glass transition temperature, and the Arrhenius equation, at a temperature significantly higher than the glass transition temperature.
3) The section on isoconversional methods (section 5.5) is merely a summary of very well-known methods, without providing any novel insights.
In conclusion, it appears that the present article does not seem to provide a summary and synthesis of any new information that could of interest to polymer researchers.
Author Response
Institute of Polymer and Dye Technology
Technical University of Lodz
90-924 Lodz, ul Stefanowskiego 12/16, Poland
Tel.: +48 42 631 32 23, Fax: +48 42 636 25 43
August 20, 2020
Materials— Open Access Journal
Dear Professor,
We are resubmitting our revised paper entitled “Lifetime Prediction Methods for Degradable Polymeric Materials – A Short Review ” by with a request to reconsider it for publication in " Materials”.We have carefully considered the Reviewers' comments. The manuscript was revised exactly according to these comments. The list of responses to the reviewer’s comments and corrections made in the manuscript is attached. In the manuscript, the changes made based on the Reviewers' comments are marked in red.
The manuscript has not been previously published, is not currently submitted for review to any other journal, and will not be submitted elsewhere before a decision is made by this journal.
For correspondence please use the following information:
corresponding author: Anna Masek
Institute of Polymer and Dye Technology
Technical University of Lodz
90-924 Lodz, ul Stefanowskiego 12/16, Poland
Tel.: +48 42 631 32 93
Fax: +48 42 636 25 43
e-mail: anna.masek@p.lodz.pl
Yours sincerely,
PhD, Dsc Anna Masek
Answers to Reviewer #1 comments
Reviewer #1: (1)The manuscript is a review of the lifetime prediction of polymeric materials. This is an important topic in the field of polymer science and the review is comprehensive in terms of the number of publications covered. The article, however, is found to be lacking in technical rigor. Of the 20 pages of the article, the first four pages are elementary polymer science. For example, even a moderately advanced reader of the article would know about the significance of polymer in the present-day life, and statements such as "it is currently difficult to find a field in which plastics are not being used" is not necessary. Similarly, a discussion of thermoplastics and thermosets is also not required.
Reviewer #1: (2) There are many statements that are unclear. For example:
2.1) Page 7: "Due to different constants of the reaction rates ki at different temperatures Ti, the same reaction threshold xa is reached after different reaction times ti." Equation (4) is cited. But there is no variable called "xa" in this equation. Furthermore, what is F(x) in Equation 4? What is the basis of the assumption that it is linearly proportional to time, ti?
The Equation 4 was incorrect: in parenthesis should be: xa (line number 279).
Where is the reaction state function.
Endlife criterion is defined as the time ti taken for a physical or chemical property to attain a threshold consistent with the function xa. It is necessary to know the trend curves for the material’s properties with test times for several experimental conditions, in order to extrapolate a useful lifetime. Only after extrapolating the results, we are able to get the graph with the linear relationship of ln(t) as a function of inverse temperature (Figure 4). Line number 303.
Assuming that Equation 4 is correct, it is not clear how the combination of Equation 3 and Equation 4 results in Equation 5. How is the variable "B," called the reaction constant, related to the parameters of equations 3 and 4.
There were 2 mistakes:
1st: the correct Equation 4 is: ;
2nd: the variable B is the constant (not reaction constant). Line number 286.
An explanation of combination of Equation 3 and 4 results in Equation 5:
Equation 3
Equation 4
After logarithming Equation 3 we obtain:
Therefore:
– constant (B).
Generally, these 3 equations have been taken from the Reference [15].
2.2) Similarly, other equations in this section (5.2 Arrhenius model) are completely unclear. First, there is a mistake in Equation 8 (dα/dt in Equation 8 should actually be dα/dT). Next, Figure 3 is not properly explained, and Equation 9 and a statement associated with it ("the correct choice of kinetic parameters can be achieved with many linear regressions") is not technically rigorous, even though it may have been copied verbatim from reference # 89.
We agree with the Reviewer's comment. In Equation 8 was mistake (line number 296). The explanation of Figure 4 (the number has been changed) has been improved. Line number 297-302. The Equation 9 and the sentence related to this equation have been deleted (line number 305-307).
2.3) In section 5.3: "The TTSP is based on the perception that the short-term and long-term behavior of viscoelastic materials is similar at higher temperatures." I am not sure what this statement means.
This sentence has been improved: This method is determined from an empirical assumption that the influence of temperature and time on the characteristics are equipollent. For a short-period accelerated aging test at higher temperature the change of behavior of characteristic variable is the same to those measured for longer period but at normal condition. Line number 323-326.
2.4) The concept of time-temperature superposition has not been properly explained. In Equation 11, why is the value of the shift factor equal to 1 at the reference temperature?
We went wrong, it was an author’s assumption in his research. In other publications we have not found any information that this value is equal 1. That’s way we decided to delete this sentence. Line number 347.
2.5) One page 10, the authors have stated that the Arrhenius equation describes the behavior below the glass transition temperature, and the WLF equation is applicable above the glass transition temperature. However, WLF equation is generally found to be applicable close to the glass transition temperature, and the Arrhenius equation, at a temperature significantly higher than the glass transition temperature.
We agree with the Reviewer's comment. This sentence has been improved. Line number 362-363.
Reviewer #1: (3) The section on isoconversional methods (section 5.5) is merely a summary of very well-known methods, without providing any novel insights.
In the section 5.5 we added the part related to Ozawa–Flynn–Wall corrected method by N. Sbirrazzuoli et al. (line number 454-463) and we also described an advanced isoconversional method by S. Vyazovkin, which is an alternative calculation method of activation energy (line number 478-491).
Reviewer 2 Report
In this minireview the authors described the methods to be utilized to predict the lifetime of polymers.
Although this is a very important concept that should be highlighted more in the scientific literature, the scope of the manuscript is not focused and clear. For example, in the title the authors stress “the significance of..” but this aspect have never been fully discussed and highlighted in the main text.
In the abstract the authors use the term “biodegradable”, while in the main text they use degradable and refer exclusively to thermal degradation of commodity polymers.
Hence, it is suggested to specify to which type of polymers these considerations are addressed and under which conditions. Indeed, the focus is merely on the extrapolation of data from induced thermal degradation. If so, the title, abstract and introduction should be narrowed down and the authors should clear select what to discuss.
Particular attention should indeed be paid to the terminology used, as already said above the word biodegradable/degradable should be selected with special care; polymer “destruction” at page 7 should be replace with a more suitable terminology.
The authors should also acknowledge that degradable polymers degradable while thermal processing and this can of course influence the predictions of the life time.
Moreover, it is suggested to support the description of all the models discussed with examples of polymers to which they have been practically applied to and graphs to show the reader how to operate.
A more appropriate refer 94 is also suggested, since the current example refer to honey.
Author Response
Institute of Polymer and Dye Technology
Technical University of Lodz
90-924 Lodz, ul Stefanowskiego 12/16, Poland
Tel.: +48 42 631 32 23, Fax: +48 42 636 25 43
August 20, 2020
Materials— Open Access Journal
Dear Professor,
We are resubmitting our revised paper entitled “Lifetime Prediction Methods for Degradable Polymeric Materials – A Short Review ” by with a request to reconsider it for publication in " Materials”.We have carefully considered the Reviewers' comments. The manuscript was revised exactly according to these comments. The list of responses to the reviewer’s comments and corrections made in the manuscript is attached. In the manuscript, the changes made based on the Reviewers' comments are marked in red.
The manuscript has not been previously published, is not currently submitted for review to any other journal, and will not be submitted elsewhere before a decision is made by this journal.
For correspondence please use the following information:
corresponding author: Anna Masek
Institute of Polymer and Dye Technology
Technical University of Lodz
90-924 Lodz, ul Stefanowskiego 12/16, Poland
Tel.: +48 42 631 32 93
Fax: +48 42 636 25 43
e-mail: anna.masek@p.lodz.pl
Yours sincerely,
PhD, Dsc Anna Masek
Answers to Reviewer #2 comments
Reviewer #2: In this minireview the authors described the methods to be utilized to predict the lifetime of polymers.
Reviewer #2: (1) Although this is a very important concept that should be highlighted more in the scientific literature, the scope of the manuscript is not focused and clear. For example, in the title the authors stress “the significance of..” but this aspect have never been fully discussed and highlighted in the main text.
We agree with the Reviewer's comment. In the section 4. Accelerated aging of polymeric materials it has been improved (line number 214-219). But also we decided to change the title of this paper: Lifetime Prediction Methods for Degradable Polymeric Materials – A Short Review. Line number 2-4.
Reviewer #2: (2) In the abstract the authors use the term “biodegradable”, while in the main text they use degradable and refer exclusively to thermal degradation of commodity polymers.
We agree with the Reviewer's comment. This part has been replaced with the following sentence: “An accurate prediction of the service life of a material is very important in terms of safety, especially considering elastomeric construction materials (e.g. aircraft components, defense applications, nuclear reactor safety construction components)”. Line number 14-17. We have been defined which type of materials are the foundation of this paper in title, abstract and introduction.
Reviewer #2: (3) Hence, it is suggested to specify to which type of polymers these considerations are addressed and under which conditions. Indeed, the focus is merely on the extrapolation of data from induced thermal degradation. If so, the title, abstract and introduction should be narrowed down and the authors should clear select what to discuss.
It has been improved. This paper is concentrated on degradable polymers and methods of lifetime prediction using accelerated thermal ageing tests and extrapolation of data from induced thermal degradation. Line number 22-23; 69-70. The title of this paper also has been changed.
Reviewer #2: (4) Particular attention should indeed be paid to the terminology used, as already said above the word biodegradable/degradable should be selected with special care; polymer “destruction” at page 7 should be replace with a more suitable terminology.
These sentences have been improved (line number 260). All parts related to biodegradable polymers have been removed (line number 17-19; 61-63).
Reviewer #2: (5) The authors should also acknowledge that degradable polymers degradable while thermal processing and this can of course influence the predictions of the life time.
We agree with the Reviewer's comment. This sentence has been added in Conclusions (line number 501-503).
Reviewer #2: (6) Moreover, it is suggested to support the description of all the models discussed with examples of polymers to which they have been practically applied to and graphs to show the reader how to operate.
In each section related to kinetic models, the examples of application in literature have been presented (line number 313-317; 350-353; 378-381; 424-429; 450-452; 475-477; 480-481). We added 2 more graphs in section 5.2 and 5.3 (line number 282 and 337) but for other methods the specific data are necessary. Our next aim is to do the research (accelerated aging of various polymers) and predict the lifetime using the different kinetic models. We are going to create the correlation between the methods described in this paper.
Reviewer #2: (7) A more appropriate refer 94 is also suggested, since the current example refer to honey.
We agree with the Reviewer's comment. This refer has been changed (line number 358).
Reviewer 3 Report
The manuscript reviews significance and various approaches to predict the lifetime of polymeric materials. The authors can improve its quality by:
1) Citing the literature examples in further detail. Only two references in the article mention the names of the contributing scientists which is not a traditional style of a review article.
2) Providing more advanced level details in section 2.3 which focuses on the basic principles of thermal degradation of polymer degradation which are available on every college textbook.
3) Including more figures and improving the quality (font size, resolution and clarity) of existing figures.
Author Response
Institute of Polymer and Dye Technology
Technical University of Lodz
90-924 Lodz, ul Stefanowskiego 12/16, Poland
Tel.: +48 42 631 32 23, Fax: +48 42 636 25 43
August 20, 2020
Materials— Open Access Journal
Dear Professor,
We are resubmitting our revised paper entitled “Lifetime Prediction Methods for Degradable Polymeric Materials – A Short Review ” by with a request to reconsider it for publication in " Materials”.We have carefully considered the Reviewers' comments. The manuscript was revised exactly according to these comments. The list of responses to the reviewer’s comments and corrections made in the manuscript is attached. In the manuscript, the changes made based on the Reviewers' comments are marked in red.
The manuscript has not been previously published, is not currently submitted for review to any other journal, and will not be submitted elsewhere before a decision is made by this journal.
For correspondence please use the following information:
corresponding author: Anna Masek
Institute of Polymer and Dye Technology
Technical University of Lodz
90-924 Lodz, ul Stefanowskiego 12/16, Poland
Tel.: +48 42 631 32 93
Fax: +48 42 636 25 43
e-mail: anna.masek@p.lodz.pl
Yours sincerely,
PhD, Dsc Anna Masek
Answers to Reviewer #3 comments
Reviewer #3: The manuscript reviews significance and various approaches to predict the lifetime of polymeric materials. The authors can improve its quality by:
Reviewer #3: (1) Citing the literature examples in further detail. Only two references in the article mention the names of the contributing scientists which is not a traditional style of a review article.
We agree with the Reviewer's comment. We improved the style of citing the literature in some places (line number 313-317; 350-353; 378-381; 424-429; 450-452; 455; 475-477; 479).
Reviewer #3: (2) Providing more advanced level details in section 2.3 which focuses on the basic principles of thermal degradation of polymer degradation which are available on every college textbook.
In this section we added more information about the UV degradation of polyolefins that are the example of degradable polymers (line number 157-170). Also we added the sentence that the more detailed description of general mechanism of thermal degradation will appear in our next paper.
Reviewer #3: (3) Including more figures and improving the quality (font size, resolution and clarity) of existing figures.
We included 2 more graphs in section 5.2 and 5.3 (line number 282 and 337) but for other methods the specific data are necessary. Our next aim is to do the research (accelerated aging of various polymers) and predict the lifetime using the different kinetic models. We are going to create the correlation between the methods described in this paper. The quality of existing figures has been improved (line number 96; 110 and 303).
Round 2
Reviewer 1 Report
Most of the information provided in the manuscript can be found in standard reference books on thermal analysis of polymers. There are several books in the literature devoted to this topic. The manuscript repeats the information, without providing sufficient details, so that if may not be a useful starting point for a reader interested in applying the equations (the published reference books or monographs would serve that purpose). Rather than repeat all of the information that is already available in books, it would be more helpful if the review article focused on how the methods have been applied by researchers and what their key conclusions or inferences were. The review should be most of paragraphs such as that on page 12, line 378 (Hu et al. applied the WLF equation...) Currently, there are only a few such paragraphs.
Reviewer 2 Report
The manuscript has been improved.
Reviewer 3 Report
Good job, it reads better now!